# Bridging the Safety Gap: A Guardrail Pipeline for Trustworthy LLM Inferences

## Abstract

We present Wildflare GuardRail, a guardrail pipeline designed to enhance the safety and reliability of Large Language Model (LLM) inferences. Wildflare GuardRail integrates four key functional modules, including Safety Detector, Grounding, Customizer, and Repairer, and addresses safety challenges across multiple dimensions of LLM inferences. Wildflare GuardRail incorporates an unsafe content detection model that identifies issues such as toxicity, bias, and prompt injection, a hallucination detection model that identifies hallucinated LLM outputs and simultaneously provides explanations for the hallucinations, and a fixing model that corrects LLM outputs based on these explanations. Additionally, Wildflare GuardRail employs Grounding to enrich user queries with relevant context, and utilizes Customizerto allow users to define flexible protocols for handling specific safety requirements. Our experiments demonstrate that Wildflare GuardRail enhances safety and robustness in LLM inferences, offering adaptable and scalable solutions for LLM inferences.

## 1 Introduction

Large Language Models (LLMs) inference guardrail is critical in real-world applications. Safety risks may exist in the user prompts as well as LLM responses during the users' interactions with LLMs. Malicious users may exploit the vulnerabilities of LLMs by interacting with them with crafted prompts, e.g., prompt injections Liu et al. (2023); Kumar et al. (2023); Zhu et al. (2023); Chu et al. (2024); Tedeschi et al. (2024); Zhao et al. (2024). The outputs generated by LLMs can also be offensive, discriminatory, nonsensical, factually incorrect (e.g. hallucinations), etc Zhang et al. (2023; 2024a); Wang et al. (2023); Fan et al. (2023); Huang et al. (2023a); Xu et al. (2024b), as the models are often trained on publicly available data that have uncontrolled sources and may include unsafe contents, which can propagate through the model's responses and produce harmful or inappropriate outputs.

Safeguarding LLMs is crucial and cannot be overstated. Unsafe inputs can not only manipulate the LLM outputs but also reveal sensitive information, bypass system instructions, or even execute harmful commands Russinovich et al. (2024); Xu et al. (2024a); Chang et al. (2024); Liu et al. (2023); Kumar et al. (2023); Zhu et al. (2023); Chu et al. (2024); Tedeschi et al. (2024); Zhao et al. (2024). Problematic LLM outputs can also confuse users, perpetuate negative stereotypes, and undermine public trust in LLM applications. This is particularly problematic in use cases where the accurate response is critical, such as healthcare, finance, and legal systems.

Addressing the safety issues associated with LLM inference is complex, as safety risks can arise at any point when processing a user prompt and often necessitate coordination of different methods. To safeguard LLM inference from unsafe or problematic outputs, standalone approaches deploy ML models to detect different types of safety risks in the LLM outputs (Hanu & Unitary team, 2020; Lees et al., 2022; Markov et al., 2023). However, while these models can identify safety issues, they are not able to fix errors in the outputs. This gap can be partially bridged by employing a separate model designed specifically for fixing detected errors. Nevertheless, this approach still fails to address certain challenges, such as hallucinations, which often stem from insufficient, inaccurate, or outdated source information (Xu et al., 2024b; Zhang et al., 2023; Huang et al., 2023a). In light of this, we can employ retrieval-augmented generation (RAG) Lewis et al. (2020); Chen et al. (2024); Gao et al. (2023), which mitigates hallucinations by enriching the user query with additional

contextual information from external data sources, thus effectively enhancing the LLM's responses in more factual content. However, these approaches fail to support quick customization of user needs and cannot handle rapidly changing contexts and system settings, e.g. newly recognized offensive terms related to society or politics that require to be filtered out in the content generated by LLMs. In such cases, a simpler, code-based rule approach, denoted as "wrappers", can be more effective and flexible Mohiuddin (2024). As an example, consider a scenario where users want to add some texts at the beginning of the LLM outputs to warn if there are any phishing URLs in the texts. To achieve this goal, we can utilize crafted regex patterns to obtain URLs in the LLM outputs, and call APIs, such as Google Safe Browsing Google (2005), to find out unsafe URLs. Such a method does not require model training or additional supplementary data sources and can be easily deployed in the LLM system, thus is more flexible to be quickly adapted to new safety challenges as they emerge.

Enhancing the overall safety of LLM inference requires a comprehensive pipeline that orchestrates different functional components, such as ML models, RAG, and wrappers, while ensuring these components work collaboratively and harmoniously. As a result, developing a comprehensive guardrail pipeline for LLM inference becomes essential. A guardrail pipeline for LLM inference integrates various functional modules, which manages and mitigates potential safety risks from a global perspective while enabling users to customize workflows and wrappers to achieve high efficiency and flexibility.

This paper proposes Wildflare GuardRail, a guardrail pipeline that enhances LLM inference safety. Wildflare GuardRail systematically manages safety challenges in LLM applications comprehensively, thus advancing a safe and scalable deployment of LLM applications. It also adapts to new safety challenges with high flexibility, thus is suitable for changing user requirements and system environments. Wildflare GuardRail orchestrates several key components with different specializations, including 1) SAFETY DETECTOR that identifies safety risks in user inputs and LLM outputs, 2) GROUNDING that utilizes vector databases to contextualize user queries, 3) CUSTOMIZER that leverages lightweight wrappers to edit LLM outputs according to user needs, and 4) REPAIRER that correct errors detected in the LLM outputs, especially hallucinations.

Our contributions are summarized as follows:

*i*) **Comprehensive safety pipeline.** We propose Wildflare GuardRail, a comprehensive pipeline for safeguarding LLM inferences. Wildflare GuardRail incorporates multiple modules, including SAFETY DETECTOR, GROUNDING, CUSTOMIZER, and REPAIRER, to comprehensively address safety challenges, rather than treating each safety issue in isolation.

*ii*) **Specialized fine-tuned models.** We utilized our pretrained LLM[1] as the base model and fine-tuned three models: *i*) an unsafe content detection model that identifies issues such as toxicity, bias, prompt injection, etc (§4.1); *ii*) a hallucination detection model that simultaneously detects hallucinated content and provides explanations for the hallucinations (§4.2); and *iii*) a fixing model that corrects problematic LLM outputs based on the explanations provided by the hallucination detection model (§7). Our models are light-weighted and can be deployed on edge devices.

*iii*) **Multifaceted hallucination handling.** We address the hallucination issue through a multifaceted approach. We first detect hallucination as well as generates the explanations of hallucination with SAFETY DETECTOR(§4.2), then utilize REPAIRER to correct the problematic content based on the reason of hallucination (§7).

*iv*) **Effective grounding.** We propose two indexing methods that assist retrieving knowledge from vector data storage in GROUNDING, and compare their effectiveness in **Exp 3** (§5).

*v*) **Flexible User-defined safety protocols.** Our pipeline allows users to define protocols for editing LLM outputs or addressing emerging safety challenges through CUSTOMIZER. This approach is adaptable to evolving user requirements and supports real-time solutions for addressing safety issues in LLM deployments, without the need for pretraining or fine-tuning to meet new requirements (§6).

*Takeaways: i*) A single method is insufficient to address the wide range of potential risks in LLM inference. Different functional modules with specialized capabilities are necessary to handle various aspects of safety. *ii*) The term "AI safety" is too narrow for the current AI landscape and should be

---

[1]Details of the model will be added later to meet anonymity requirements.

expanded to "AI quality assurance" to capture broader challenges and possibilities, including safety, accuracy, reliability, and robustness, in the rapidly evolving AI environment.

## 2 RELATED WORK

Moderation-based harmfulness mitigation approaches leverage rule-based methods, ML classifiers, and human interfaces to monitor, evaluate, and manage the outputs produced by LLMs to ensure the outputs generated by LLMs are safe, appropriate, and free from harmful content (Markov et al., 2023; Rebedea et al., 2023; Lees et al., 2022; Hanu & Unitary team, 2020; Mohiuddin, 2024).

**Close-sourced solutions.** OpenAI Moderation API (Markov et al., 2023) and Perspective API (Lees et al., 2022) utilize ML classifiers to detect undesired contents. These approaches provide scores for pre-defined categories of harmful content, such as toxicity, identity attacks, insults, threats, etc. These tools are widely used in content moderation to filter out harmful content and has been incorporated into various online platforms to protect user interactions (team, 2008). However, they are less adaptable to emerging safety risks as they are not open-sourced and cannot be finetuned.

**Opensourced solutions.** LlamaGuard (Inan et al., 2023) leverages the zero/few-shot abilities of the Llama2-7B architecture (Touvron et al., 2023) and can adapt to different taxonomies and sets of guidelines for different applications and users. Despite its adaptability, LlamaGuard's reliability depends on the LLM's understanding of the categories and the model's predictive accuracy. However, deploying LlamaGuard on edge devices is challenging due to its large number of parameters, which typically exceed the computing resources available on edge devices. Detoxify (Hanu & Unitary team, 2020) offers open-source models designed to detect toxic comments. These models, based on BERT (Devlin et al., 2018) and RoBERTac (Liu et al., 2019) architectures, are trained on the Jigsaw datasets (JIGSAW, 2019; 2018; 2020). Detoxify provides pre-trained models that can be easily integrated into other systems to identify toxic content. Also, the models are able to recognize subtle nuances in language that might indicate harmful content, making them effective for moderation.

**Customizable solutions.** Guardrails (Mohiuddin, 2024) and Nvidia NeMo Guardrails (Rebedea et al., 2023) employ customizable workflows to enhance LLM safety. Guardrails (Mohiuddin, 2024) define flexible components, called "rails", to enable users to add wrappers at any stage of inference, which enables users to add structure, type, and quality guarantees to LLMs outputs. Such rails can be code-based or using ML models. However, it does not have self-developed model and miss a unified solution for general cases. Nvidia NeMo Guardrails (Rebedea et al., 2023) functions as an intermediary layer that enhances the control and safety of LLMs. This framework includes pre-implemented moderation dedicated to fact-checking, hallucination prevention, and content moderation, which offers a robust solution for enhancing LLM safety.

We compare our approaches with the existing approaches in Table 1.

Table 1: Comparison of moderation-based harmfulness mitigation approaches

| Feature | Perspective API | Open AI | Nvidia NeMo | GuardRails | Detoxify | Llama Guard | Ours |
|---|---|---|---|---|---|---|---|
| Open-sourced | ✗ | ✗ | ✓ | ✓ | ✓ | ✓ | ✓ |
| Self-developed model | ✓ | ✓ | ✗ | ✗ | ✓ | ✓ | ✓ |
| Deployable on edge devices | ✓ | ✓ | - | - | ✓ | ✗ | ✓ |
| Zero-shot generalization | ✓ | ✓ | - | - | ✓ | ✓ | ✓ |
| Explainable results | ✗ | ✗ | ✗ | ✗ | ✗ | ✗ | ✓ |
| Flexible workflow | ✗ | ✗ | ✓ | ✓ | ✗ | ✗ | ✓ |

## 3 WILDFLARE GUARDRAIL OVERVIEW

Wildflare GuardRail enhances safety of LLM inputs and outputs while improving their quality. Specifically, it achieves two goals, 1) all user inputs are safe, contextually grounded, and effectively processed, such that the inputs to the LLMs are of high-quality and informative; and 2) the output generated by the LLMs are evaluated and enhanced, such that the outputs passed to users can be both relevant and of high quality. The pipeline can be partitioned into two parts, including 1) processing before LLM inference that enhances user queries, and 2) processing after LLM inference that detects undesired content and handle them properly. We overview our pipeline in Figure 1.

Figure 1: Overview.

**1. Pre-inference processing.** Before sending user queries to LLMs, Wildflare GuardRail detects if there are any safety issues in the queries with SAFETY DETECTOR and ground the queries with context knowledge with GROUNDING. SAFETY DETECTOR monitors user inputs to identify and reject queries that might be unsafe. The monitoring includes typical safety checks, including toxicity, stereotypes, threats, obscenities, prompt injection attacks, etc. Any form of unsafe content will lead to the queries being rejected. Inputs that pass this initial safety check are grounded with context with GROUNDING, where the user query is contextualized and enhanced with relevant knowledge retrieved from the vector data storage. By equipping the query with some context knowledge, the LLM can do inference with enriched information, thus can reduce hallucinations when generating responses. The details of SAFETY DETECTOR and GROUNDING will be introduced in §4 and §5, respectively.

**2. Post-inference processing.** Upon LLM finishing inference, SAFETY DETECTOR detects safety issues in the LLM outputs, specifically, hallucinations. This is because LLM applications typically leverages well-developed LLMs or APIs, such as LLaMA (Touvron et al., 2023) and ChatGPT API (Markov et al., 2023), which are generally safe and less likely to generate toxic or other unsafe content, while hallucinations occur frequently. The SAFETY DETECTOR identifies hallucinations and provides reasons for the hallucinations, such that Wildflare GuardRail can utilize the reasoning for later refinement of the LLM outputs. To achieve goal, Wildflare GuardRail employs a text generation model to generate explainable results, and adjusts the loss function during training to ensure the model to produce classification results. After SAFETY DETECTOR finishes detection, REPAIRER fixes the problematic content or aligns the outputs with some rule-based wrappers to meet user expectations. If the outputs are difficult to fix, such as those containing hallucinations, REPAIRER will call a fixing model to fix the answers. Details about REPAIRER can be found in §7.

Table 2: Training data for unsafe content detection in SAFETY DETECTOR

| Dataset | Train | Validation | Test |
|---|---|---|---|
| HEx-PHI (Qi et al., 2024) | 330 | 0 | 0 |
| OpenAI (Markov et al., 2023) | 160 | 1,500 | 0 |
| Hotpot QA (Yang et al., 2018) | 3,000 | 2,500 | 500 |
| Truthful QA (Lin et al., 2021) | 500 | 100 | 100 |
| Awesome ChatGPT Prompts (Akın, 2023) | 0 | 150 | 0 |
| Jigsaw Unintended-Bias Data (JIGSAW, 2019) | 100,000 | 2,000 | 300 |
| GPT-Jailbreak (Romero, 2023) | 0 | 78 | 0 |
| Jailbreak Hao (2023) | 400 | 0 | 70 |
| Personalization Prompt (Siah, 2024) | 1,000 | 800 | 200 |
| QA Chat Prompts (Testing, 2024) | 0 | 200 | 0 |
| ChatGPT Prompts (Rashad, 2023) | 350 | 0 | 0 |
| 10k Prompts Ranked (Community, 2024a) | 500 | 500 | 200 |
| Iterative Prompt (Community, 2024b) | 500 | 500 | 200 |
| Instruction Following (Kojohnjaratkul, 2023) | 200 | 340 | 0 |

## 4 WILDFLARE GUARDRAIL SAFETY DETECTOR

SAFETY DETECTOR addresses unsafe inputs and inappropriate LLM responses to ensure that both the user queries provided to the models and the LLM outputs are safe and free from misinformation.

---

**Algorithm 1:** Training data processing for hallucination detection.

---

**Inputs:** $\mathcal{D}$: a training dataset that contains "context", "inputs", "llm_answer", and "labels" for hallucination; *prompt_template*: for formulating the hallucination detection data, see Figure 2; *GPT_reasoning_template*: for generating prompts for GPT API, see Figure 2.
**Outputs:** $\mathcal{D}_t$: the training dataset.

**1 Function** $process\_data(\mathcal{D})$ **begin**
**2**     $\mathcal{D}_t \leftarrow \phi$
**3**     **for** $d \in \mathcal{D}$ **do**
**4**        **if** *is_hallucination(d)* **then**
**5**           $halu\_reason \leftarrow GPT\_API(GPT\_reasoning\_template(d["question"],$
            $d["context"], d["llm\_answer"]))$
**6**           $response \leftarrow$ "Yes, " $+ halu\_reason$
**7**           $d' \leftarrow prompt\_template(d["question"], d["context"], d["llm\_answer"], response)$
       **else**
**8**           $d' \leftarrow prompt\_template(d["question"], d["context"], d["llm\_answer"], "No.")$
**9**        $\mathcal{D}_t \leftarrow d'$
**10**     **return** $\mathcal{D}_t$

---

## 4.1 Unsafe Input Detection

We developed a model to detect unsafe contents in user queries before they are processed by LLMs for inference. While existing approaches categorize unsafe content into various types (e.g., toxicity, prompt injection, stereotypes, harassment, threats, identity attacks, and violence) (Markov et al., 2023; Wang et al., 2023; Hanu & Unitary team, 2020), our method employs a unified, binary classification model finetuned based on our opensourced LLM (Anonymous, 202X), classifying content as safe or unsafe.

This strategy offers several key advantages, as follows: *i*) By fine-tuning our base model, which has been trained on vast amounts of data, the classification model can leverage pre-existing knowledge relevant to safety detection. *ii*) A binary classification of "safe" and "unsafe" is both efficient and sufficient for LLM services, as any unsafe query should be rejected, regardless of the specific risk. *iii*) This approach avoids the complexities and potential inaccuracies of categorizing overlapping or ambiguous types of unsafe content in some publicly available datasets. For example, toxicity toward minority groups could also be classified as bias, but current datasets may inadequately capture such nuances. *iv*) Using straightforward code logic, we can transform public datasets for safety detection into clear safe/unsafe labels, minimizing ambiguity and ensuring high-quality training data.

The biggest challenge in training such model is the discrepancy between the training data and real-world user query distributions, where using traditional datasets alone can result in poor performance due to their divergence from actual user queries (Markov et al., 2023). To mitigate these issues, we integrated data of various domains and contexts to better simulate the variety of unsafe queries that users might submit. We crafted a training dataset by combining samples randomly selected from 15 public datasets, as summarized in Table 2. Such a dataset captures diverse contents in user inputs in practice, thus can be more representative on potential real-world inputs.

## 4.2 Output Hallucination Detection and Reasoning

Hallucinations occur when the LLM generates responses that is inaccurate, fabricated, or irrelevant (Filippova, 2020; Maynez et al., 2020; Huang et al., 2023b; Rawte et al., 2023). Despite appearing coherent and plausible, hallucinated LLM responses are unreliable, often containing fabricated, misleading information that is divergent from the user input, thus fail to meet users' expectations and severely undermine the trustworthiness and utility of the LLM applications. While grounding can mitigate hallucinations by contextualizing user inputs and enriching the informativeness of user queries, it cannot eliminate hallucinations entirely. This is because hallucinations stem from nearly every aspects of LLM training and inference, such as low-quality training data (Lin et al., 2021;

Figure 2: Prompt templates and sample training data for hallucination detection and reasoning.

---

**Algorithm 2:** Inference with hallucination detection model.

**Inputs:** $\mathcal{M}$: hallucination detection model; $tokenizer$: tokenizer for $\mathcal{M}$; $q$: a query submitted by users; $context$: the context to answer the question; retrieved from vector data storage; $a$: the answer returned by an LLM for the question; $inference\_prompt\_template$: see Figure 2.

1 **Function** $inference(\mathcal{M}, q, context, a, k)$ **begin**
2    $prompt \leftarrow inference\_prompt\_template(q, context, a)$
3    $tokenized\_prompt \leftarrow tokenizer(prompt)$
4    $halu\_res \leftarrow \mathcal{M}.generate(tokenized\_prompt)$
5    $first\_word\_logits \leftarrow halu\_res.logits[0],$
6    $results \leftarrow softmax(first\_word\_logits)$
7    $top\_k\_probs, top\_k\_indices \leftarrow top(results, k)$
8    $P_{halu}(a) \leftarrow compute\_halu\_prob(top\_k\_probs, top\_k\_indices)$ // See Definition 10
9    **if** $P_{halu}(a) \geq 0.5$ **then return** True;
10    **return** False

---

Kang & Choi, 2023) and randomness of sampling strategies (Chuang et al., 2023), and moreover, the very nature probabilistic properties of LLMs.

Effectively handling hallucinations in LLM responses is both crucial and challenging for producing high-quality LLM responses. Existing works that detect presence of hallucinations are insufficient (Manakul et al., 2023; Liu et al., 2021). To provide high-quality responses to users, we should handle the detected hallucinations properly, i.e., obtaining the explanations for the hallucinations in the LLM responses and further, fixing the hallucinated responses if possible.

To this end, we propose utilizing our own LLM as base model (Anonymous, 202X) to finetune a hallucination detection model for detecting hallucinated content and providing explanations, and further, facilitating the subsequent REPAIRER in §7. The design of the model has the following advantages: *i) classification*: it identifies the presence of hallucinations in the LLM output; and *ii) reasoning*: it generates explanations for the hallucinated contents, offering insights for the subsequent correction in REPAIRER; *iii) simultaneous classification and reasoning*: it process *i)* and *ii)* at the same time, which saves computation cost and improves efficiency; and *iv) vast pre-training data*: it leverages pre-existing knowledge on hallucination in the base model, which may potentially benefit hallucination detection and reasoning.

**Training.** We fed our base model with hallucination dataset to train a model for both detecting and reasoning for the hallucination. However, public available datasets for hallucinated LLM responses are mainly classification datasets with texts and labels, e.g., HaluEval (Li et al., 2023a). To address this, we utilize the GPT4 API (Markov et al., 2023) to generate explanations for hallucinated contents, and define a prompt template to create structured prompts based on the classification data to make it suitable for classification and reasoning simultaneously. We demonstrate the prompt templates and sample training data in Figure 2, and summarize data processing in Algorithm 1.

**Inference.** We expect the LLM to directly output results whether the LLM response contains hallucinations, *i.e.*, the first token of outputs to be "Yes" or "No" as detection results, according to the formatted data sample in Figure 2. However, the first token of the LLM response is probabilistic due to the self-autoregressive nature of decoder-based text generation LLMs. To obtain desired outputs, we formulate the text-generation outputs by utilizing the top-$k$ first tokens (and their possibilities) of the outputs to generate classification results. By default, $k$ is 10.

**Definition 1** (Probability of hallucination). *Let $a$ be an LLM answer, let $\{t_1, ..., t_k\}$ be the top-k potential first token, and let $\{p_1, ..., p_k\}$ be their top-k probabilities. Let $T$ be a tokenization function, and let $T("Yes")$ and $T("No")$ be the tokens corresponding to "Yes" and "No", respectively. The probability of hallucination in $a$ is $P_{halu}(a) = \frac{\sum_{i=1}^{k} P(t_i | t_i \in T("Yes"))}{\sum_{i=1}^{k} P(t_i | t_i \in T("Yes")) + \sum_{i=1}^{k} P(t_i | t_i \in T("No"))}$*

Detection results with $P_{halu}(*) \geq 0.5$ indicate the content is classified as "hallucinated"; otherwise, the content is "safe". The detailed procedure of inference is described in Algorithm 2.

# 5 WILDFLARE GUARDRAIL GROUNDING

Wildflare GuardRail GROUNDING enhances the contextual richness and informativeness of user queries by leveraging external knowledge stored in vector database. Thus, LLMs can utilize such contextual knowledge to generate high-quality outputs, particularly by grounding user queries before they are passed to the LLMs for inference.

To support similarity search over the knowledge data, Wildflare GuardRail creates vector indexes by vectorizing plaintext knowledge. This involves vectorizing entire knowledge entries to create vector indexes. Wildflare GuardRail employs two primary methods for indexing: *i*) *Whole Knowledge Index* that creates indexes based on each entire data entry in the datasets; and *ii*) *Key Information Index* that indexes only the key information in each data entry, i.e., questions in QA datasets. Whole Knowledge Index reflects the data distribution and ensurers that the indexed data captures the contextual variety and complexity found in real-world queries, while Key Information Index focuses on the core information of each data entry, thus facilitates efficient retrieval of relevant data. We evaluate the effectiveness of indexes with *callback*, i.e., the probability of successfully retrieving the original records from a dataset using Top-$k$ queries. We experimentally evaluate the indexing methods in §8.

**Definition 2** (Callback). *Let $D_v$ be a vector data storage that contains $n$ records, let $Q$ be a plaintext user query set, and let $I(Q)$ be the vector index created based on $Q$. For each query $q \in Q$, let $I_q$ be the vector index created based on $q$, and let $D_v(I_q)$ denote the set of Top-k records returned by querying $D_v$ with $I(q)$, and let $r_q$ denote the most relevant record of $q$ in $D_v$. The callback for Top-k queries on the query set $Q$ is defined as:*

$$C_k(Q) = \frac{1}{|Q|} \sum_{q \in Q} [r_q \in D_v(I_q)]$$

*where $[\cdot]$ is Iverson Bracket Notation (Iverson, 1962), equal to 1 if the condition inside is true, and 0 otherwise.*

To ensure effective and informative grounding, the distribution of the index should closely align with query patterns, i.e., query distributions. By grounding user queries with knowledge retrieved with a proper index, the LLMs can generate contextually appropriate responses, and further, reduce hallucinations and improve the quality of the responses.

# 6 WILDFLARE GUARDRAIL CUSTOMIZER

Wildflare GuardRail CUSTOMIZER utilizes lightweight wrappers to flexibly edit or customize LLM outputs to fix some small errors or enhancing the format of the answer. The wrappers integrate code-based rules, APIs, web searches, and small models to efficiently handle editing and customization tasks according to user-defined protocols. Wildflare GuardRail CUSTOMIZER offers several key advantages. It facilitates rapid development and deployment of user-defined protocols, which crucial in production environments where real-time adjustments are necessary. In scenarios where training

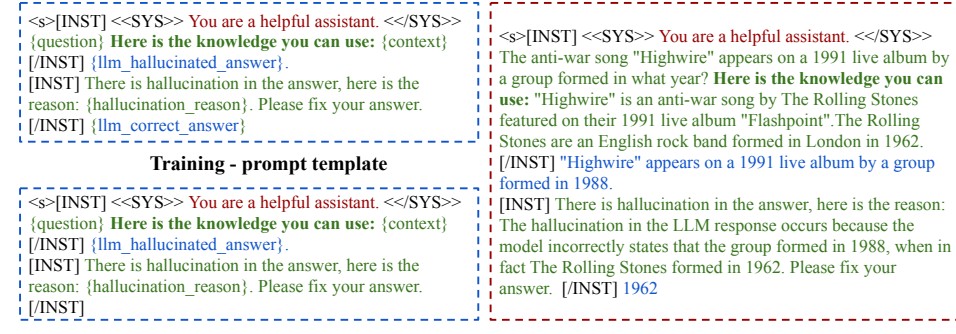

Figure 3: Prompt templates and sample training data for REPAIRER.

or fine-tuning LLMs is unfeasible due to time or resource constraints, this method provides an alternative for immediate output customization. Moreover, the wrappers enable flexible incorporation of various tools and data sources, which enhances the applicability of Wildflare GuardRail and reduces resource-intensive LLM calls.

**Example 1** (Warning URLs). *The objective was to detect if LLM outputs contain URLs and prepend a warning message of the unsafe URLs at the beginning of the LLM outputs. CUSTOMIZER should check the safety of the URLs founded, i.e., whether they are malicious or unreachable, and includes such information in the warning if they were unsafe. CUSTOMIZER utilizes a regular expression pattern to identify URLs within the text. Upon URLs founded, CUSTOMIZER calls APIs for detecting phishing URLs, such as Google SafeBrowsing (Google, 2005), and assess the accessibility of the benign URL by issuing web requests. Malicious URLs, as well as unreachable URLs that return status codes of 4XX, are added in the warning at the beginning of the LLM outputs.*

Note that the task in Example 1 cannot be achieved through prompt engineering when querying LLMs, as the warning must appear at the beginning, and LLMs generate content token by token, making later content unpredictable. We use an example to show this feature in Appendix A and experimentally evaluate the efficiency of CUSTOMIZER wrappers in **Exp 4** in §8.

## 7 WILDFLARE GUARDRAIL REPAIRER

Wildflare GuardRail REPAIRER addresses errors in the LLM outputs that are challenging to resolve through editing with wrappers in CUSTOMIZER, particularly, hallucinated content. REPAIRER analyzes and corrects the hallucinated output based on the reason for the hallucinations generated by the hallucination detection model.

Wildflare GuardRail REPAIRER takes several key inputs, including the user's original query, the context retrieved with GROUNDING, the hallucinated responses generated by the LLM, as well as the reason for hallucination. Given these inputs, REPAIRER corrects the flawed output according to the hallucination reason. To enable REPAIRER to handle hallucinations effectively, we leverage the same hallucination detection dataset as SAFETY DETECTOR, i.e., HaluEval (Li et al., 2023a), that contains user questions, contexts, hallucinated LLM answers, and correct answers. We also designed a customized data template that incorporates the information. The data templates for training, inference, as well as an example for the training data, are demonstrated in 3.

## 8 EXPERIMENTS

**Experimental Setting.** We use our self-developed model as our base model for finetuning safety detection models. SAFETY DETECTOR model is trained with 15 datasets, as descrioded in Table 2. Hallucination detection model is trained with HaluEval dataset (Li et al., 2023a). GROUNDING experiments are conducted on the following datasets: E-Commerce Tian (2023), ChatDoctor Li et al. (2023b), PatientDoctor Dey (2023a). The information of all datasets for experiments is summarized

in Table 4 in Appendix §B. Evaluations and model training experiments are conducted on a server with 8 NVIDIA H100 GPUs.

**Exp 1. Unsafe user inputs detection in SAFETY DETECTOR.** The datasets and the number of records involved in training, validation, and test phases are summarized in Table 2. We compare our approach with Detoxify-Roberta (Hanu & Unitary team, 2020), Detoxify-BERT (Hanu & Unitary team, 2020), Nvidia NeMo GuardRail (Rebedea et al., 2023), OpenAI Moderate (Markov et al., 2023), and PerspectiveAPI (Lees et al., 2022) in Figure 4. Results show that our model achieves comparable performance with OpenAI API. Overall, our model demonstrates robust performance across key metrics, indicates its effectiveness and reliability in real-world applications.

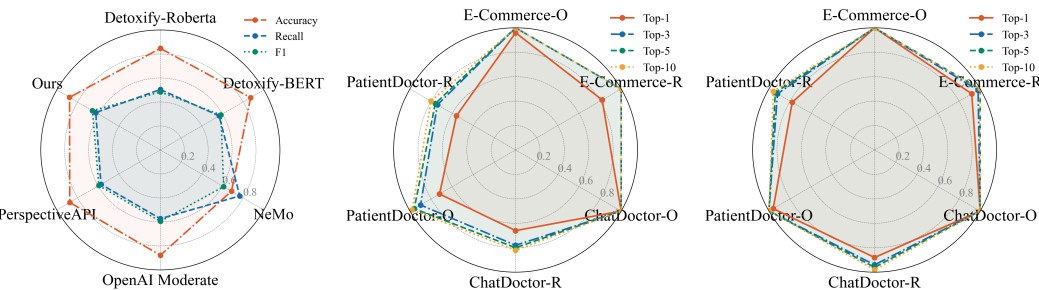

Figure 4: Safety detection    Figure 5: Whole index    Figure 6: Key index

**Exp 2. Hallucination detection in LLM outputs in SAFETY DETECTOR.** We fine-tuned our hallucination detection model using the HaluEval dataset Li et al. (2023a). For each subset, we utilized 8,000 data samples for training, 1,500 for validation, and 500 for testing. Our model achieved an accuracy of 0.78 on the testing dataset.

**Exp 3. Evaluation of different indexing methods in GROUNDING.** We leveraged *callback* to evaluate the effectiveness of the two indexing methods in GROUNDING. We utilized datasets that contain important knowledge for evaluation, where inaccurate retrieval can cause financial losses or harmful medical advice. We selected three QA datasets, including E-Commerce dataset Tian (2023) that contains customer service interactions on an online platform, and two healthcare datasets, PatientDoctor dataset Dey (2023a) and the ChatDoctor dataset Li et al. (2023b), which contain QA pairs between doctors and patients. To comprehensively evaluate retrieval performance and simulate user queries real-world applications, we used two types of queries, including i) *original queries* that match original questions in the datasets ("O" in Figure 1 and Figure 6), and ii) *rephrased queries* generated with language models (i.e., TinyLlama (Zhang et al., 2024b) or a summarization model (Falconsai, 2023)) based on the original questions to simulate variability in user questions ("R" in Figure 1 and Figure 6). For each evaluation, we randomly selected 50 questions from the dataset to form a question set $Q$, and processed Top-$k$ queries to compute a callback $C_k(Q)$, where $k$ is set to 1, 3, 5, and 10. We recorded the callbacks for Whole Knowledge Index and Key Information Index in Figure 5 and Figure 6, respectively. The results indicate that Key Information Indexing outperformed Whole Knowledge Indexing, as key information indexes reflects the user queries better. Also, both original queries and rephrased queries achieved high callback rates, which demonstrates the effectiveness of vector retrieval when handling varied user inputs.

Table 3: URL Detection Task

| Metrics | Ours | TinyLLama | Mistral-7B | LLama2-7B | LLama3-8B | Falcon-40B |
|---|---|---|---|---|---|---|
| Avg. Time (s) | **1.06** | 13.17 | 10.93 | 9.10 | 20.10 | 34.67 |
| Detection Acc. | **100.00%** | ✗ (Fail) | 91.67% | 83.33% | 37.50% | ✗ (Fail) |
| Validation Acc. | **83.33%** | ✗ (Fail) | 45.83% | 54.17% | 37.50% | ✗ Fail |

**Exp 4. Efficiency of wrappers in CUSTOMIZER.** We evaluated the efficiency of CUSTOMIZER in with the URL detection and validation task in Example 1 in §6. We randomly selected 15 records from the each of the E-Commerce dataset (Tian, 2023) and the RedditSYACURL Dataset (Dey, 2023b), combined each record to construct texts that contained URLs, and set 20% probability of

inserting some malicious URLs into the text. In implementation, we leveraged Regex pattern for detecting URLs, Google SafeBrowsing (Google, 2005) for detecting malicious URLs, and sent HTTP requests to the safe URLs to verify their reachability. We compared CUSTOMIZER with several models, including TinyLLama (Zhang et al., 2024b), Mistral-7B (Jiang et al., 2023), LLama2-7B (Touvron et al., 2023), and Falcon-40B (Almazrouei et al., 2023). The results are shown in Table 3. We record average time to process one query, the success rate of detecting URLs (Detection Acc.), and the accuracy of identifying unsafe URLs (Validation Acc.). The results show that Wildflare GuardRail CUSTOMIZER takes much less time (1.06s per query) and significantly outperforms calling the models for editing LLM outputs. Also, TinyLLama and Falcon-40B failed to detect any URLs in the contents. Though Mistral is able to detect URLs with a high accuracy of 91.67%, the accuracy of identifying unsafe URLs is only 45.83%.

**Exp 5. Effectiveness of fixing hallucinations in REPAIRER.** We fine-tuned our fixing model using the HaluEval dataset Li et al. (2023a). We selected the QA and dialogue subsets. For each subset, we utilized 8,000 data samples for training, 1,000 for validation, and 1000 for testing. We utilize Vectara hallucination detection model (Vectara, 2023) for evaluating the consistency between the LLM outputs and the information provided in the original data, including the user questions, the contexts, and the correct answers. We utilized the 100 records in the test dataset of the HaluEval-QA dataset for evaluation. Results show that our fixing model improves the quality of the LLM outputs by a lot. Moreover, 80.7% of the hallucinated data were fixed using REPAIRER.

## 9 CONCLUSION

This paper introduces Wildflare GuardRail, a pipeline designed to enhance the safety and reliability of LLM inferences. By integrating several key components, including SAFETY DETECTOR, GROUNDING, and CUSTOMIZER, and REPAIRER, we addressed critical challenges such as unsafe inputs, hallucination, contextually grounding, output customization. We believe that our work can serve as a stepping stone to contribute to the advancement of the field of LLM safety.

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

## A    EXAMPLE OF AN ERRONEOUS RESPONSE

We use an example to demonstrate that some tasks cannot be achieved through prompt engineering when querying LLMs, as LLMs generate content token by token, making it impossible to predict subsequent tokens at the beginning of generation. We use the following prompt to ask GPT4:

```
Write an English poem about a rabbit; please include
information at the beginning of the poem about how
many times the word "rabbit" appears in the poem.
```

Below is the response, which claims the word "rabbit" appears 4 times. However, "rabbit" appears 5 times in total. This showcases the necessity of including post-inference wrappers to fix simple errors in the LLM outputs.

```
(The word "rabbit" appears 4 times in this poem.)

In a field where soft grasses sway and dip, A rabbit
hopped beneath the moon's silver whip.  With ears like
twin sails in the cool night air, Each twitch and turn
done with utmost care.
This rabbit, cloaked in twilight's gentle hug, Nestled
in shadows, a cozy, earthen snug.  Dreams of dandelion
fields in sunny sprawl, Where no shadows loom, nor
fears enthrall.
With a nose that twitches at secrets untold, And paws
that dance over leaves of gold, The rabbit finds joy
in the simplest of things-- The warmth of the sun, the
freedom of springs.
But as dawn breaks and stars bid goodbye, Back to the
burrow, this rabbit must fly.  For the world is wide
and wolves roam free, Yet in his burrow, safe as can
be, this rabbit shall be.
```

## B    SUMMARY OF DATASETS

We describe the dataset we utilized in experiments in Table 4.

Table 4: Dataset Descriptions

| Task | Dataset | Data size | Description |
|------|---------|-----------|-------------|
| Unsafe Content Detection | HEx-PHI (Qi et al., 2024) | 330 | This dataset contains 330 harmful instructions (30 examples x 11 prohibited categories) for LLM harmfulness evaluation. |
| | OpenAI (Markov et al., 2023) | 1680 | OpenAI moderation dataset that contains prompts that are annotated with their own taxonomy. |
| | Hotpot QA (Yang et al., 2018) | 113k | QA pairs based on Wikipedia knowledge. |
| | Truthful QA (Lin et al., 2021) | 827 | A QA dataset that contains 817 questions spanning 38 categories, including health, law, finance, and politics. |
| | Awesome GPT Prompts (Akın, 2023) | 153 | Awesome prompt examples to be used with ChatGPT. |
| | Jigsaw Unintended-Bias Data (JIGSAW, 2019) | 2M | Comment data that contains labels for unsafe content. |
| | GPT-Jailbreak (Romero, 2023) | 79 | ChatGPT jailbreak prompts. |
| | Jailbreak (Hao, 2023) | 1.3k | A dataset that contains jailbreak prompts and benign prompts. |
| | Personalization Prompt (Siah, 2024) | 10.4k | Prompt-response pairs for personalized interactions with LLMs. |
| | QA-Chat Prompts (Testing, 2024) | 200 | A QA dataset. |
| | ChatGPT Prompts (Rashad, 2023) | 360 | A dataset that contains human prompts and ChatGPT responses. |
| | 10k-Prompts Ranked (Community, 2024a) | 10.3k | A dataset of prompts with quality rankings created by 314 members of the open-source ML community using Argilla, an open-source tool to label data. |
| | Iterative Prompt (Community, 2024b) | 20k | A dataset of user prompts. |
| | Instruction Following (Kojohnjaratkul, 2023) | 514 | An instruction dataset. |
| | ToxicChat (Lin et al., 2023) | 10165 | [Evaluation data in **Exp1**] A dataset for toxicity detection. |
| Hallucination Detection | HaluEval (Li et al., 2023a) | 10k | A dataset for evaluating hallucinations. We utilize the "qa" subset that contains dialogue histories, contexts (knowledge), right responses, and hallucinated responses. |
| Vector Retrieval Evaluation | E-Commerce (Tian, 2023) | 65 | We use the "faq" subset that contains QA pairs between users and service agents. |
| | PatientDoctor (Dey, 2023a) | 379k | Dialogue data between doctors and patients. |
| | ChatDoctor dataset (Li et al., 2023b) | 119.4 | Dialogue data between doctors and patients. |
| Repairere Wrapper Evaluation | E-Commerce dataset (Tian, 2023) | 1.89k | We use the "faq" subset (as described above) and the "product" subset that contains descriptions of products. |
| | RedditSYACURL Dataset (Dey, 2023b) | 8.61k | A dataset that contains titles, summaries, and links of articles. |

