# OpenReview forum: "Bridging the Safety Gap: A Guardrail Pipeline for Trustworthy LLM Inferences"
_ICLR.cc/2025/Conference — ICLR 2025 Conference Withdrawn Submission_

### Official Review · Reviewer_LDJ1 · 2024-10-20

**Soundness:** 2
**Presentation:** 1
**Contribution:** 1
**Rating:** 3
**Confidence:** 3

**Summary:**

This paper proposes Wildflare GuardRail, a guardrail pipeline to enhance the LLMs' safety and reliability. Wildflare GuardRail integrates safety detector, grounding, customizer, and repairer, which offers a comprehensive safety pipeline.

**Strengths:**

1. The author proposes Wildflare GuardRail, a safety guardrail pipeline that can effectively improve the model's safety and reliability.
2. The author developed Wildflare GuardRail with their own model, which can have better accuracy, efficiency, and flexibility.

**Weaknesses:**

1. Compared to the related work like Llamagurad, the advantage claimed by this work is vague.  To be more specific, in Table 1, there are several metrics that are unnecessary, including "Deployable on edge devices" (The author did not provide details on how they evaluate this metric, also, I believe that Llamaguard can be developed in the edge devices with techniques like pruning or quantization ([Li et al, 2024](https://proceedings.mlsys.org/paper_files/paper/2024/file/42a452cbafa9dd64e9ba4aa95cc1ef21-Paper-Conference.pdf)), "Explainable Results" (Is it necessary to explain the results as long as the guard model can fix the misaligned output?). Authors should provide more explanations or experiment details to prove that these metrics matter when evaluating the effectiveness of the guardrails.
2. The experiment results showed in Figure 4, 5, 6 also cannot show the advantage of Wildflare GuardRail. In figure 4, it achieves a similar performance to the other works. In Figure 5 and Figure 6, there lack of baseline comparisons. The author should also provide a comparison with other works and report the results in Figure 5 and Figure 6. Based on my understanding, "GROUNDING" has no difference with existing RAG models ([Lewis et al, 2020](https://proceedings.neurips.cc/paper/2020/file/6b493230205f780e1bc26945df7481e5-Paper.pdf), [Min et al, 2024](https://arxiv.org/pdf/2308.04430)). The author should provide a comparison with these models.
2. Many details for data preparation, model training, and evaluation are missing, including:
    1. The basic information of the "self-trained" model. What is the architecture? What is the number of parameters? How it is trained? All this information is missing and the author should provide more details.
     2. The author does not mention the experiment's details on randomness analysis, including the sampling strategies they are using, how many repetitions for each experiment. The author should report the confidence intervals for all results to show the randomness.
     3. In line 193, "the outputs are difficult to fix" lacks a clear definition, author should provide more explanation on when the repairer will call a fixing model to fix the answer
     4. In line 219, the definition of "labels for hallucination" is missing. How do we obtain this label for each example? Is it included in the original training dataset, or is it needed to be labeled during the filtering process?
     5. In Line 257, "randomly selected from 15 public datasets" is vague. How many examples are randomly sampled from each dataset? Is each dataset evenly sampled? Did the author filter the identical examples? I also did not find the number of examples for the final training dataset.
     6. In line 330, why there are a set of tokens that correspond to Yes and No? I assume there is only one token that corresponds to "Yes" (tokenid=9642 in Llama3) and one token that corresponds to "No" (tokenid=2822 in Llama3)? Is the author referring to the tokens like  " Yes"(tokenid=7566 in Llama3) and " No" (tokenid=2360 in Llama3)? If so, what tokens are included when computing the probability? Also, is there a possibility that none of the candidate tokens are in the top-k (10 in paper) tokens? Why don't we just determine based on the logit order of the candidate tokens? The author should provide more details on how they compute the probability of hallucinations.
     7. In line 455, when training the hallucination detector, the author segment the HaluEval dataset as (train, val, test)=(8000, 1500, 500), but when training the repairer, the author segments the HaluEval datasets as (train, val, test) = (8000, 1000, 1000). Any explanation on this?
3. The role of the safety detector is somewhat misleading. In Figure 1, the author shows that the safety detector works for both "unsafe input detection" and "output hallucination detection", but based on the descriptions in the later paragraph, it looks like the safety detector and hallucination detector are two modules that are fine-tuned with different strategies (See line 429-430). The author should provide further clarification on this.
4. Based on my understanding, the hallucination detector's success is based on its capability to factual knowledge. The author only tested the performance of the hallucination detector on the same HaluEval dataset (in-distribution dataset). I am curious about its performance on the out-of-domain data and the author should provide this ablation study if possible.
5. In Line 247, the claim "A binary classification of “safe” and “unsafe” is both efficient and sufficient for LLM services" is misleading. There are a lot of discussions on the definition of AI safety and security ([Qi, et al 2024](https://arxiv.org/pdf/2405.19524)) and it is not that simple. Different countries have different policies and definitions regarding what is allowed and what is not allowed for AI generation. The author should provide more details or proofs to show the sufficiency of binary classification.

**Questions:**

I have listed my questions in the weaknesses section.

---

### Official Review · Reviewer_nh9e · 2024-10-28

**Soundness:** 1
**Presentation:** 3
**Contribution:** 1
**Rating:** 3
**Confidence:** 4

**Summary:**

Wildflare GuardRail is a pipeline designed to improve the safety and reliability of Large Language Model (LLM) inferences. It consists of four modules: SAFETY DETECTOR, GROUNDING, CUSTOMIZER, and REPAIRER. These modules address safety challenges across various LLM inference dimensions. Wildflare GuardRail includes an unsafe content detection model, hallucination detection model, and fixing model. It also uses GROUNDING to enrich user queries and CUSTOMIZER to define flexible protocols.

**Strengths:**

+ The paper is easy to read.

**Weaknesses:**

- The contributions of the work are minor, and it is like a combination of existing techniques.
- Many technical details are missing.
- The difference between customizer and repairer is unclear.

**Questions:**

1. The biggest problem of the manuscript is that the contributions of the work are limited. Detection, grounding, customizer, and repairer are all fast-developed. What the authors do is more like combining different procedures into a pipeline, which is not novel.

2. Section 4.1: what only randomly picks a small portion of the data as the training data for detection? It seems like there is more happening here.

3. Section 4.2: What are the advantages of using one unified model for multiple tasks, compared to using other open-source models for specific areas? Moreover, why use "Yes" or "No" as the hallucination detection results?

4. Sections 6 & 7: Many technical details are missing here. How to correct the flawed output? Why not detailed introduce your models?

---

### Official Review · Reviewer_gX1e · 2024-10-31

**Soundness:** 2
**Presentation:** 2
**Contribution:** 2
**Rating:** 3
**Confidence:** 3

**Summary:**

The authors propose a guardrail pipeline called **Wildflare GuardRail** to improve the reliability of LLM (Large Language Model) responses during inference. Wildflare consists of a safety detector, grounding module, hallucination detector, customizer, and repairer. The process of Wildflare GuardRail can be described as follows:
1. For a given user query, the safety detector identifies potential safety issues, such as harmful content or jailbreak attempts. If deemed unsafe, the query is rejected.
2. The query, along with relevant context (provided by the grounding module), is then fed into the LLM.
3. After receiving the LLM's response, the hallucination detector checks for potential hallucinations and offers explanations, considering user-defined protocols set in the customizer.
4. Based on the explanation, the repairer corrects the output and generates a refined response.

The authors conducted experiments to evaluate each component of the pipeline. Results indicate that the proposed method outperforms baseline approaches.

**Strengths:**

1. The paper is well-organized and clearly written.
2. The systematic guardrail pipeline proposed can reduce toxic content and hallucinations in LLM responses.
3. Experimental results demonstrate the effectiveness of Wildflare GuardRail.

**Weaknesses:**

1. The paper lacks novelty. The pipeline is essentially a combination of existing techniques (e.g., safety and hallucination detection modules).
2. The contribution is not entirely clear. For instance, while the authors claim that the specialized fine-tuned models represent a contribution, the training of such models on existing datasets, as in the case of the safety detector, seems rather straightforward.
3. The training data for the hallucination detector includes GPT-4's annotations and explanations, which may introduce additional hallucinations, raising concerns about the reliability of the detector.
4. In **Experiment 5** (Line 497), the authors use another hallucination detection model (i.e., Vectara, from prior work) to evaluate the repairer's fix rate. This choice seems less rigorous; at least one manual evaluation should have been included.

**Questions:**

1. Safety detection, grounding, hallucination detection, and repair seem to be distinct tasks. Could the authors further justify the necessity of addressing all these issues within a single system?
2. For each module, can the authors provide one positive and one negative example? This would help enhance my understanding of the work.

---

### Official Review · Reviewer_aspC · 2024-11-02

**Soundness:** 2
**Presentation:** 2
**Contribution:** 2
**Rating:** 3
**Confidence:** 4

**Summary:**

This paper introduces the "Wildflare GuardRail," a pipeline for enhancing safety and LLMs from the pre-inference to post-inference stage.  The pipeline consists of four main components: 1) SAFETY DETECTOR that identifies safety risks in user inputs and LLM outputs,  such as toxicity, bias, prompt injection, and hallucinations. 2) GROUNDING that utilizes vector databases to contextualize user queries, 3) CUSTOMIZER that uses lightweight wrappers to edit LMs according to user needs, and 4)REPAIRER that correct errors detected in the LLM outputs, focusing on hallucinations.

The paper’s contribution: introducing a multi-module pipeline that enhances LLM safety; develops open-source models for safety, hallucination moderation as well as more flexible customization for users.

**Strengths:**

This paper considers breaking down the pre-inference to post-inference life cycle into different modules, and comes up with techniques to improve safety and reliability for each step. The various modules are also flexible and adaptable. The paper provides an open-source attempt to replicate the safety/ reliability ecosystem that proprietary models like ChatGPT may have in place. The paper structure is clear and experiments are easy to follow.

**Weaknesses:**

- Safety: The training set is not a full/ realistic representation of real world jailbreak attempts. The safety detection comparison should also be made against some safety-specific models, such as LlamaGuard. Simply grouping all of the risky behaviors under one umbrella “unsafe” category could also lead to over-refusal issues.
- Hallucination: Hallucination detection only uses 1 dataset for training, which is definitely not adequate for properly learning this task. The classifier is thus very limited to the scope of tasks covered in the training dataset. More importantly, the judge model for hallucination needs to have the ground truth for reference in order to properly achieve the task. However, this is not a reasonable assumption, as no model is hallucination free and knows the ground truth, especially if the user query is out of distribution. In addition, the paper gives few details on hallucination evaluation, and the reported detection accuracy is relatively low (0.78).
- Grounding: this requires creating vector indices for the entire database (for each specific task), which is not a reasonable assumption, especially for general-purpose LMs. It also does not capture more general retrieval settings where useful information could be coming from different sources.
- Customizer: It doesn’t seem reasonable to assume that open source pipelines can be equipped with such API access to process all incoming prompts. Moreover, it involves clearly defining which tools to use for what purposes, for which the authors do not discuss the practicality in actual use case. The authors claim that the wrapper system is lightweight, but there is little reporting on the real-time deployment performance (such as time delay) and the scalability of such approach.
- Repairer: Similar to the point regarding the hallucination module, the reliability of the
- There is very limited quantitative evaluation of performance along the various axes the model is evaluated on. Some details such as API tool use are missing. Also no supplementary material to further validate or replicate some of their results.
- Lack of novelty within each part of the pipeline. Although the whole framework is a While the Wildflare GuardRail pipeline offers an original integration of existing methods, the individual components themselves—such as content filtering, grounding, and hallucination repair—are not novel on their own.

**Questions:**

- Can you provide more assessments and numbers on safety evaluation? What is the distribution of synthetic vs. real-world jailbreaking prompts in the dataset being collected? What about the over-refusal rate on tasks like “how to kill a python program”
- For the hallucination module, could you explain the dataset choice and also provide more quantitative results such as false positive and false negatives? Some important metrics worth reporting include false positive, false negative etc.
- Certain types of hallucinations—such as those with factual grounding errors in complex or domain-specific contexts—may not be easily detectable or repairable by current modules. Would like to see some ablations on these type of tasks.
- For the URL experiment comparison with other models like Mistral and TinyLLama, are all models equipped with the same API call access? How are they being evaluated on?
- Would appreciate addressing the comments in the weakness section as well.

---

### Note · Authors · 2024-11-13

I have read and agree with the venue's withdrawal policy on behalf of myself and my co-authors.